# Radiomics and Artificial Intelligence in Uterine Sarcomas: A Systematic Review

**DOI:** 10.3390/jpm11111179

**Published:** 2021-11-11

**Authors:** Gloria Ravegnini, Martina Ferioli, Alessio Giuseppe Morganti, Lidia Strigari, Maria Abbondanza Pantaleo, Margherita Nannini, Antonio De Leo, Eugenia De Crescenzo, Manuela Coe, Alessandra De Palma, Pierandrea De Iaco, Stefania Rizzo, Anna Myriam Perrone

**Affiliations:** 1Department of Pharmacy and Biotechnology, University of Bologna, 40126 Bologna, Italy; gloria.ravegnini2@unibo.it; 2Radiation Oncology, IRCCS Azienda Ospedaliero-Universitaria di Bologna, 40138 Bologna, Italy; martina.ferioli4@unibo.it (M.F.); alessio.morganti2@unibo.it (A.G.M.); 3Department of Experimental, Diagnostic and Specialty Medicine, University of Bologna, 40138 Bologna, Italy; antonio.deleo@unibo.it; 4Medical Physics Unit, IRCCS Azienda Ospedaliero-Universitaria di Bologna, 40138 Bologna, Italy; lidia.strigari@aosp.bo.it; 5Division of Oncology, IRCCS—Azienda Ospedaliero Universitaria di Bologna, 40138 Bologna, Italy; maria.pantaleo@unibo.it (M.A.P.); margherita.nannini@aosp.bo.it (M.N.); 6Division of Oncologic Gynecology, IRCCS-Azienda Ospedaliero-Universitaria di Bologna, 40138 Bologna, Italy; eugeniadecrescenzo@gmail.com (E.D.C.); pierandrea.deiaco@unibo.it (P.D.I.); 7Department of Medical and Surgical Sciences (DIMEC)-Centro di Studio e Ricerca delle Neoplasie Ginecologiche (CSR), University of Bologna, 40138 Bologna, Italy; 8Department of Radiology, IRCCS Azienda Ospedaliero-Universitaria di Bologna, 40138 Bologna, Italy; manuela.coe@aosp.bo.it; 9Forensic Medicine and Integrated Risk Management Unit, Azienda Ospedaliero-Universitaria di Bologna, via Albertoni 15, 40138 Bologna, Italy; alessandra.depalma@aosp.bo.it; 10Istituto di Imaging della Svizzera Italiana (IIMSI), Ente Ospedaliero Cantonale (EOC), Via Tesserete 46, 6900 Lugano, Switzerland; stefania.rizzo@eoc.ch; 11Facoltà di Scienze biomediche, Università della Svizzera italiana (USI), via Buffi 13, 6900 Lugano, Switzerland

**Keywords:** uterine tumors, uterine sarcoma, fibroids, radiomics, artificial intelligence, deep learning, machine learning

## Abstract

Background: Recently, artificial intelligence (AI) with computerized imaging analysis is attracting the attention of clinicians, in particular for its potential applications in improving cancer diagnosis. This review aims to investigate the contribution of radiomics and AI on the radiological preoperative assessment of patients with uterine sarcomas (USs). Methods: Our literature review involved a systematic search conducted in the last ten years about diagnosis, staging and treatments with radiomics and AI in USs. The protocol was drafted according to the systematic review and meta-analysis preferred reporting project (PRISMA-P) and was registered in the PROSPERO database (CRD42021253535). Results: The initial search identified 754 articles; of these, six papers responded to the characteristics required for the revision and were included in the final analysis. The predominant technique tested was magnetic resonance imaging. The analyzed studies revealed that even though sometimes complex models included AI-related algorithms, they are still too complex for translation into clinical practice. Furthermore, since these results are extracted by retrospective series and do not include external validations, currently it is hard to predict the chances of their application in different study groups. Conclusion: To date, insufficient evidence supports the benefit of radiomics in USs. Nevertheless, this field is promising but the quality of studies should be a priority in these new technologies.

## 1. Introduction

Uterine body tumours (UBTs) are represented by endometrial carcinomas (ECs) and sarcomas (USs). ECs are the most common female cancers of the reproductive system in high-income countries, with a favourable prognosis in most patients [1,2]. On the contrary, USs are rare and among the most lethal gynaecological cancers [3].The clinical management of UBTs is complicated by the tumour heterogeneity and by the difficult classification both in terms of histological types and risk classes. Hence, UBTs require a detailed assessment of multiple variables, including, but not limited to, clinical, radiological, pathological and genomic parameters, to achieve the risk stratification needed to plan the treatment. Unfortunately, the assessment of most of these parameters is operator-dependent and therefore potentially affected by inaccuracies even by experienced operators. Moreover, the need to include different parameters into the risk assessment, each associated with some risk of error, amplifies the likelihood of incorrect prognostic stratification. This issue is of particular importance in ECs where risk stratification, as reported by the European Society of Medical Oncology (ESMO)-risk, is based almost entirely on parameters that are difficult to reproduce, in particular histological type and degree of differentiation [4]. Furthermore, these issues are even more evident in high-grade ECs, and the integration between various risk factors (histopathological and molecular) is nowadays an open question [5]. With regard to the USs, the problem is even more complex. The paucity of parameters useful for risk stratification is worsened by the lack of accurate imaging criteria able to differentiate, before surgery, USs from their benign counterparts (fibroids) [6]. Indeed, the histological examination of the surgical specimen is the only way to reach a definitive diagnosis. There are still some unsolved problems for certain borderline tumours, such as atypical fibroids, where it is difficult to classify individual cases between benign and malignant categories. Nevertheless, the use of fertility-sparing interventions, minimally invasive surgical techniques or technically inadequate resections, such as in cases of intraoperative tumour fragmentation (morcellation), can dramatically impact both quality of life and prognosis [7]. Therefore, the technical details of surgical resection should be also considered in US risk stratification. Based on this background, US treatment lacks a personalized approach and the opportunities of precision medicine. However, in the near future an increasing number of data deriving from the various -*omics* will be increasingly available also in these types of tumours. The potential refinement of risk stratification systems resulting from this new scenario will require the ability to manage and analyse large databases. To this purpose, analyses based on artificial intelligence (AI) systems could be able to overcome the human cognitive possibilities and therefore are considered very attractive [8,9]. Over the last decade there has been an increasing focus on AI methods applied to medicine and, particularly, to oncology. The main reasons of this growing interest are the advantages of personalised medicine based on predictive models developed through the analysis of large databases; an additional reason is the possibility to standardize the evaluation of several parameters (e.g., histopathological and radiological) based on an automated assessment. Furthermore, awareness is growing due to the fact that tumours and their response rates, during and after treatment, greatly differ between specific cancers and inter-patients and, thus, different adaptive strategies are required to optimize cancer control and minimize toxicity [10,11]. Therefore, during the clinical pathways, data collected in medical records and radiological images are used to generate a flow of information (dataflow) which reconstructs the natural history of the disease. All these considerations together allow us to shed light on all the possible nuances of each patient’s tumour characteristics. All this information represents an increasingly important body of data in the scientific literature and is the basis for the construction of artificial intelligence algorithms. The ultimate goal of this process is to help physicians to shape a personalized view of the patient before and during the treatment process and then to guide medical decisions [12].

Given the clinical management issues of USs, AI and more specifically radiomics could promote a more efficient identification of new biomarkers and new diagnostic and prognostic criteria, playing a key role in improving the currently available prognostic stratification systems and developing new ones.

Therefore, the purpose of this systematic review was to assess the state of the art of imaging-based AI techniques (including radiomics) applied to USs.

## 2. Materials and Methods

### 2.1. Eligibility Criteria

The PICOS framework (population, intervention, comparison, outcomes, study design) was used to formulate the questions for this study: (1) patients with malignant uterine sarcomas (population), (2) assessed with radiomics/AI (interventions), (3) and/or with standard radiological exams (comparisons), (4) diagnosis, and/or prognosis (outcomes), and (5) all types of cohort studies, including randomized controlled trials, case series and case reports (study design). The focused question was “What are the potential contributing factors on diagnosis and prognosis of radiomics/AI compared to standard radiological imaging in malignant uterine sarcomas?”. They were included in this review if they met the PICOS criteria.

### 2.2. Information Source and Search Strategies

The protocol was drafted according to the systematic review and meta-analysis preferred reporting project (PRISMA-P) [13]. The protocol was registered in the PROSPERO international register on 6 June 2021 (CRD42021253535) [14]. Our literature review involved a systematic search conducted on 20 October 2021. PubMed, Scopus, and Cochrane Library databases were systematically searched for original articles analysing the impact of radiological imaging-based AI techniques on uterine sarcomas.

### 2.3. Study Selection and Data Extraction

Relevant studies were selected using the Boolean combination of the key terms reported in Appendix A. Additionally, the reference list of reviews, meta-analyses, and all selected papers were hand-searched to acquire further relevant studies missed from the initial electronic search. Eligible studies included retrospective and prospective studies, case series and clinical trials. Exclusion criteria were as follows: preclinical studies, duplicate data, study protocols, systematic or narrative reviews, meta-analyses, letter commentaries, editorials, planning studies, imaging studies, surveys, guidelines and recommendations. After removing duplicate studies, four independent investigators (GR: biologist and expert in basic cancer research, MF: radiation oncologist and expert in gynecological cancers, SR: radiologist and expert in radiomics analyses and AMP: gynaecological oncologist and expert in clinical research) carefully read the titles and abstracts of the relevant articles and judged their eligibility. Then, the entire text of potentially eligible studies was evaluated to assess the appropriateness for inclusion. Disagreements were resolved by consensus. Potentially relevant papers were screened using title and abstract and articles that did not meet the inclusion criteria were excluded. After screening titles and abstracts, the articles were archived in a reference management system to eliminate duplicates. Subsequently, the remaining full text articles were retrieved and examined by four authors (GR, MF, SR and AMP) to independently extract data.

For each eligible article, the following data were sought and recorded: the study characteristics (first author, publication year, objective and endpoint, study design); the patients’ characteristics (cancer type, number of patients, median age, stage according to the International Federation of Gynaecology and Obstetrics, treatment setting (at first diagnosis or after recurrence)); methods applied (imaging technique, presence of a validation group, type of segmentation, model construction based on radiomics or other AI techniques, inclusion of clinical features in the AI-based model).

Finally, extracted data were crosschecked by the four investigators (GR, MF, SR and AMP).

### 2.4. Assessment of Methodological Quality

The same authors (GR, MF, SR and AMP) independently assessed the methodological quality of the selected studies. In cases of disagreement, they attempted to reach a consensus, and if they failed a senior author made the final decision. The overall quality of the included studies was critically evaluated based on the revised “Quality Assessment of Diagnostic Accuracy Studies” tool (QUADAS-2) [15]. This tool comprises four domains (patient selection, index test, reference standard and flow and timing); each domain was assessed in terms of risk of bias and a graph was constructed accordingly.

## 3. Results

### 3.1. Literature Search

The initial search yielded 754 articles. Ten studies were duplicates across PubMed and Scopus and thus eliminated, resulting in 744 studies to be screened. According to the previously described inclusion and exclusion criteria, 722 papers were excluded, and 6 full-text articles were included in this systematic review [16,17,18,19,20,21]. Details about the literature search results are reported in Figure 1.

Given the small number of the included papers and the heterogeneity of the available data, a meta-analysis to calculate the pooled results was not performed.

As shown in Table 1, the selected articles were published between 2018 and 2020. All studies were retrospective, and the number of included patients ranged between 58 and 80. The mean/median age of patients ranged between 42.1 and 58.7 years, while none of the authors reported the FIGO stage. All articles included only patients at first diagnosis.

### 3.2. Technical Aspects of the Included Studies

As shown in Table 2, all studies were based on MRI (one included both PET and MRI). A validation cohort was not included in any study. The segmentation was performed manually in all papers. The presented predictive models were based on radiomics in 2/10 and on machine learning (ML) in 4/10.

### 3.3. Quality Assessment

The overall quality assessment of the included studies is reported in Figure 2. For all domains, the risk of bias was ≤70%. The domain with the most frequent risk of bias was “reference standard”.

### 3.4. Main Findings

The main findings are reported in Table 3.

### 3.5. Lesion Characterization: Differentiation between Leiomyomas and Sarcomas

Clinical factors correlated with a diagnosis of USs included older age [17,19], interrupted endometrial cavity and ill-defined tumour margins [17].

Although a complex algorithm showed 100% sensitivity, specificity and accuracy in the differentiation of myomas from leiomyosarcomas, this classifier was deemed too complicated for routine clinical practice [16,20]. The models based on radiomics features extracted from the whole uterus outperformed the ones based on features extracted only from the macroscopic tumour or from the tumour and a small region of the surrounding tissue [18]. Furthermore, AI-based performance of multiparametric MRI was superior to PET in diagnosis, whereas MRI perfusion parameters were not helpful in differentiating benign from malignant lesions.

Finally, MRI-extracted AI-based methods were comparable to [17] or more accurate than the interpretation of experienced radiologists [19,21].

## 4. Discussion

This systematic review assessed the state of the art of imaging-based techniques (including radiomics and other AI-related imaging modalities) applied to USs. A few studies have shown that sometimes complex models, including AI-related algorithms, showed excellent accuracy in this setting [20]. However, these models are still considered too complicated for prompt inclusion in clinical practice. Moreover, being that most findings derive from retrospective series and with missing external validations, it is difficult to evaluate the generalizability of the reported results.

Our literature analysis showed a progressively growing interest in AI models in USs in recent years, including all studies published in the last two years. However, these studies are retrospective, and the lack of standardized protocols makes them very heterogeneous in terms of type of samples, of analysis and of segmentation. Moreover, the number of patients analysed is too small and the follow-up duration is too short to carry out reliable assessments.

For all these reasons, it is a shared opinion of the authors that, currently, AI models cannot be used in clinical practice to solve the problems of differential diagnosis and risk stratification in USs.

More recently, another systematic review on radiomics applied to uterine tumours was published [22]. However, that analysis focused only on ECs and not, as in our case, on USs. Nevertheless, even the authors of this latest review concluded that the available evidence is not of a sufficient level to allow the clinical application of radiomics to ECs. The main issue about USs faced by the studies included in our analysis was the dilemma of pre-operative differential diagnosis between benign and malignant lesions. Regarding the imaging techniques, MRI was the most widely used. Moreover, MRI proved to be superior to PET. However, experienced radiologists were at least equivalent to AI models in all cases and sometimes they were more accurate in the diagnostic phase. It should be noted that the selection of investigated cases was sometimes unclear. Most importantly, in some studies, different histologic types, including carcinosarcomas, were erroneously analysed together [19,21]. Furthermore, different US types, such as leiomyosarcomas and sarcomas of the endometrial stroma, were analysed together, even if they are considered to be two distinct tumours. In fact, though these cancers are classified as “sarcoma”, they have different behaviours with a diverse clinical presentation, prognosis and treatment. This bias, together with the small sample size, could have influenced the final findings of the studies.

As in radiomic studies on other organs, different strategies have also been adopted for the uterus in terms of inclusion in the analysis of the whole organ or of the macroscopic tumour only [23]. Indeed, the studies included in our review reported better results in cases of whole uterus segmentation compared to that of the tumour alone [18]. Anyway, the segmentation of the whole uterus guarantees the complete inclusion of the entire tumour sites, especially in PET examinations where the tumour edges can be poorly defined.

One of the main limitations of our systematic review is the “time factor”. In fact, given the interest in this topic, it is possible that additional studies have been published after our literature search and therefore have not been included in our analysis. The main limitations of our analysis were: (i) the small size and the heterogeneity of the included studies which obviously affects the levels of evidence of the review results; (ii) the low number of included studies; (iii) the wide range of inclusion criteria used to select patients in the analysed series that hindered the achievement of clear results. Moreover, none of the included articles provided an independent validation of the developed AI models, with obvious limits to their generalizability and thus to their external applicability. Finally, although multicentric and prospective studies are required to accurately assess the impact of AI on clinical outcomes, no analyses have been yet published.

One of the strengths of our systematic review is the involvement of a multidisciplinary team of authors, including gynaecologists, radiologists, radiation oncologists, medical oncologists and experts in basic cancer research. Indeed, team members assessed the studies in detail, each based on their expertise and knowledge. Furthermore, this systematic review provides a comprehensive overview of radiomics and AI analyses currently used, alone or in combination with other dataflow, in order to build predictive models for US diagnosis and risk stratification. Finally, this analysis could improve the shared knowledge among different specialists involved in gynaecological oncology and could pave the way for future studies on USs.

These future studies could have the purpose of: (i) defining the imaging methods in the USs, or their combinations, most appropriate for the development of AI models; (ii) evaluate the usefulness of integrated predictive models, including imaging, clinical, and molecular data; (iii) identify the most effective AI systems to produce reliable predictive models for diagnosis and risk stratification.

In conclusion, the improvement of research quality should be the future focus in this field. a multidisciplinary approach could probably avoid several biases in patients’ selection and monitoring. Despite the lack of enough evidence on obvious advantages of radiomics and more generally of AI in USs at the moment, some preliminary data suggest a potential advantage from integrating these methods with human intelligence. Further studies are needed to refine AI techniques to enable their future use in the complex clinical management of USs.

## Figures and Tables

**Figure 1 jpm-11-01179-f001:**
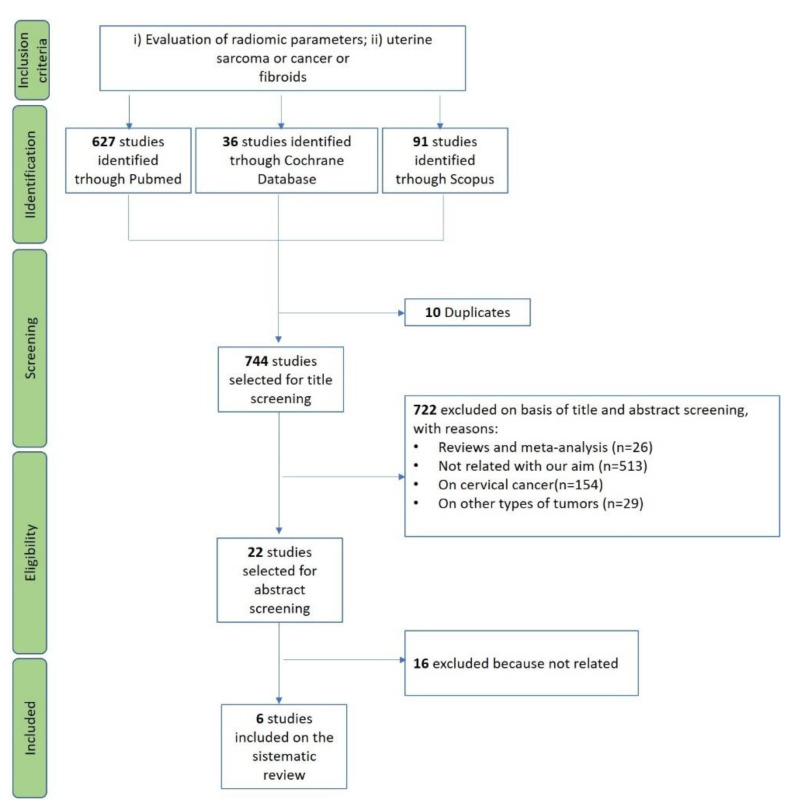
PRISMA flow-chart for the selection of studies.

**Figure 2 jpm-11-01179-f002:**
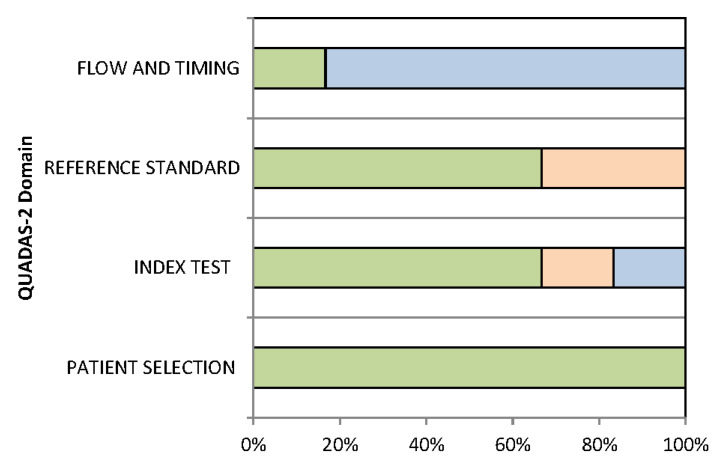
Proportion of studies with low (green), high (light orange) or unclear (light blue) RISK of BIAS.

**Table 1 jpm-11-01179-t001:** Studies included in the systematic review.

Authors	Year	Objective	Endpoint	Study Design	Cancer Type	N Patients	Mean/Median Age	FIGO * Stage	First Diagnosis or Recurrence
Malek [16]	2020	Lesion characterization	Differentiation between leiomyoma and sarcoma	Retrospective	Sarcoma and leiomyoma	65	42.1	ND	First diagnosis
Xie [17]	2019	Lesion characterization	Differentiation between leiomyoma and sarcoma	Retrospective	Sarcoma and leiomyoma	58	58.7	ND	First diagnosis
Xie [18]	2019	Lesion characterization	Differentiation between leiomyoma and sarcoma	Retrospective	Sarcoma and leiomyoma	78	ND	ND	First diagnosis
Nakagawa [19]	2019	Lesion characterization	Differentiation between leiomyoma and sarcoma	Retrospective	Sarcoma and leiomyoma	80	50.2	ND	First diagnosis
Malek [20]	2018	Lesion characterization	Differentiation between leiomyoma and sarcoma	Retrospective	Sarcoma and leiomyoma	60	44.7	ND	First diagnosis
Nakagawa [21]	2018	Lesion characterization	Differentiation between leiomyoma and sarcoma	Retrospective	Sarcoma and leiomyoma	67	54.4	ND	First diagnosis

FIGO *: International Federation of Gynaecology and Obstetrics; ND: Not declared.

**Table 2 jpm-11-01179-t002:** Methodological and technical aspects of the included studies.

Authors	Imaging Technique	Validation Group	Segmentation	Model Construction	Inclusion of Clinical Features in the Model
Malek [16]	MRI	No	Manual	ML	No
Xie [17]	MRI	No	Manual	Radiomics	Yes
Xie [18]	MRI	No	Manual	Radiomics	No
Nakagawa [19]	MRI	No	Manual	ML	No
Malek [20]	MRI	No	Manual	ML	No
Nakagawa [21]	MRI; PET	No	Manual	ML	No

ML: machine learning; MRI: magnetic resonance imaging; ND: not determinate; PET: positron emission tomography.

**Table 3 jpm-11-01179-t003:** Main findings of the selected papers.

Authors	Significant Results for Lesion Characterization: Differentiation between Leiomyomas and Sarcomas
Malek [16]	A simple algorithm showed 96.2% accuracy, 100% sensitivity and 95% specificity. The complex algorithm yielded accuracy, sensitivity and specificity of 100%. However, the complex one is more time-consuming and needs difficult imaging calculations.
Xie [17]	Ill-defined tumour margin and interrupted uterine endometrial cavity of older women were predictors of uterine sarcoma. The optimal radiomic model showed comparable efficacy with experienced radiologists.
Xie [18]	Radiomic model based on features extracted from VOI that covered the whole uterus (compared to VOI including the sole tumour or the tumour and a small piece of surrounding tissue) showed the best diagnostic performance.
Nakagawa [19]	Age was the most important factor for differentiation (*p* < 0.001). The AUC for the machine learning method used outperformed experienced radiologists in the differentiation of uterine sarcomas from leiomyomas.
Malek [20]	No perfusion parameter was able to differentiate leiomyomas from sarcomas. When the information provided by the extracted features was aggregated using a ML method, a promising discriminative power was obtained.
Nakagawa [21]	The diagnostic performance of the ML method using mp-MRI was superior to PET and comparable to that of experienced radiologists

AUC: Area under the ROC curve; ML: machine learning; MRI: magnetic resonance imaging; PET: positron emission tomography; VOI: volume of interest.

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
