# Peer review of "Radiomics and Artificial Intelligence in Uterine Sarcomas: A Systematic Review"

_jpm, 2021, doi:10.3390/jpm11111179_

Round 1
Reviewer 1 Report
This is an interesting systematic review assessing the present status of imaging techniques for verfication of uterine malignant tumors with the use of AI modeling algorithms.
The title is well difined. The abstract is compact but informative. The introduction part is well discribed giving readers a wide overview concerning the problems with uterine tumors stratification and risk validation altough this passage should be more consist.
The target of the research is well defined. The methodology of data extraction and presentation is correct and worthy to note that multidisciplinary team of researches has been created to minimalize the bias reffered to subjective nature of reviewing process.
The discussion is adequate with the results, still staying critical to their heterogenic and retrospective nature. In the conclusion it is underlined that at the moment there are no evidences for the common use of AI models in the uterine tumors predicting but this promising path may develop in the future.
There are some spelling mistakes, especially in the Figure 1, to be corrected.
Author Response
Thank you for your positive appreciations and your interesting comments. We corrected the spelling mistakes in Figure 1 and we checked the paper correcting grammatical errors. Moreover, we identified and corrected several typos.
Reviewer 2 Report
There are a few changes that are necessary regarding the use of English language:
the sentence that starts at line 32 has unclear wording in line 33
the same about the sentence starting at line 279
the sentence at line 283 needs reformulating
The study represents a good effort as there is growing interest in the medical use of AI and radiomics. But the research establishes too broad a scope and concludes with the acknowledgement of its shortfalls through the heterogeneity and the small number of the included studies and being limited by the time factor. In the end this work fails provide any meaningful results. Maybe its foundations can be used after restricting the study to either ECs or USs and allowing for more studies to be pooled and further refining the design by including more technical aspects that can be used in creating a useful radiomics platform.
Author Response
Comment 1: There are a few changes that are necessary regarding the use of English language: the sentence that starts at line 32 has unclear wording in line 33, the same about the sentence starting at line 279 , the sentence at line 283 needs reformulating
Answer 1: Thank you for your suggestions. We checked spelling mistakes and unclear sentences and we corrected them. Moreover, we identified and corrected several typos.
Comment 2: The study represents a good effort as there is growing interest in the medical use of AI and radiomics. But the research establishes too broad a scope and concludes with the acknowledgement of its shortfalls through the heterogeneity and the small number of the included studies and being limited by the time factor. In the end this work fails provide any meaningful results. Maybe its foundations can be used after restricting the study to either ECs or USs and allowing for more studies to be pooled and further refining the design by including more technical aspects that can be used in creating a useful radiomics platform.
Answer 2: Thank you for your interesting comments. We accepted your suggestion and we focused our analysis only on uterine sarcomas. Moreover, based on your comments, we checked the literature but we did not find new papers related to this topic. The small amount of available studies on artificial intelligence in uterine sarcomas, could be a starting point to deepen this topic more and more by analyzing multiple features that can allow us to provide improvements of clinical practice. Specifically, the entire manuscript was edited to focus on uterine sarcomas. The title has also been changed to: "Radiomics and artificial intelligence in uterine sarcomas: a systematic review."
Round 2
Reviewer 2 Report
Radiomics, AI and ML are undoubtedly well placed in the future of diagnostic imaging. This paper is well focused on uterine sarcomas challenging and difficult early diagnosis. Even if it concludes that radiomics is not yet powerful enough to make a difference its merits reside in evaluating the state of the art and establishing a foundation on witch to further investigate.